# Establishing Treatment Effectiveness in Fabry Disease: Observation-Based Recommendations for Improvement

**DOI:** 10.3390/ijms25179752

**Published:** 2024-09-09

**Authors:** Bram C. F. Veldman, Daphne H. Schoenmakers, Laura van Dussen, Mareen R. Datema, Mirjam Langeveld

**Affiliations:** 1Department of Endocrinology and Metabolism, Amsterdam Gastroenterology Endocrinology Metabolism, Amsterdam UMC Location University of Amsterdam, 1105 AZ Amsterdam, The Netherlands; b.c.f.veldman@amsterdamumc.nl (B.C.F.V.);; 2Department of Endocrinology and Metabolism, Platform “Medicine for Society”, Amsterdam UMC Location University of Amsterdam, 1105 AZ Amsterdam, The Netherlands; 3Department of Child Neurology, Amsterdam Leukodystrophy Center, Emma’s Children’s Hospital, Amsterdam UMC Location Vrije Universiteit, 1081 HV Amsterdam, The Netherlands

**Keywords:** Fabry disease, enzyme replacement therapy (ERT), chaperone therapy, treatment effectiveness, clinical outcomes

## Abstract

Fabry disease (FD, OMIM #301500) is caused by pathogenic *GLA* gene (OMIM #300644) variants, resulting in a deficiency of the α-galactosidase A enzyme with accumulation of its substrate globotriaosylceramide and its derivatives. The phenotype of FD is highly variable, with distinctive disease features and course in classical male patients but more diverse and often nonspecific features in non-classical and female patients. FD-specific therapies have been available for approximately two decades, yet establishing robust evidence for long-term effectiveness remains challenging. This review aims to identify the factors contributing to this lack of robust evidence for the treatment of FD with enzyme replacement therapy (ERT) (agalsidase-alfa and -beta and pegunigalsidase alfa) and chaperone therapy (migalastat). Major factors that have been identified are study population heterogeneity (concerning sex, age, phenotype, disease stage) and differences in study design (control groups, outcomes assessed), as well as the short duration of studies. To address these challenges, we advocate for patient matching to improve control group compatibility in future FD therapy studies. We recommend international collaboration and harmonization, facilitated by an independent FD registry. We propose a stepwise approach for evaluating the effectiveness of novel treatments, including recommendations for surrogate outcomes and required study duration.

## 1. Introduction

Fabry disease (FD) is a lysosomal storage disorder caused by pathogenic variants in the *GLA* gene with a very broad phenotypic spectrum. The classical phenotype of FD is well-defined with the onset of characteristic features, e.g., acroparesthesia, angiokeratoma, and cornea verticillata, in childhood or adolescence [1]. With advancing age, progressive organ involvement becomes evident through the presence of albuminuria and gradually deteriorating renal function, morphological and functional changes in the heart, and the development of cerebrovascular incidents [2]. Notably, this full classical FD phenotype is almost exclusively observed in male FD patients with *GLA* variants, resulting in (near) complete loss of α-galactosidase A (AGAL) enzyme activity. Clinical presentation in other FD patients is far more variable and may lack the disease-specific features. In females with *GLA* variants associated with classical FD in males, the clinical picture is likely influenced by genetic mosaicism on the tissue level [3]. These female patients often develop cardiomyopathy, with an average onset of clinical symptoms after age 40 years, but generally much later, often in the absence of other clinically significant organ involvement. Renal function remains within the normal population range in the majority of female patients [4], and the development of renal failure with FD as a single cause is rare [5]. They may be at higher risk for the development of white matter lesions in the brain, transient ischemic attack (TIA), and stroke, though the relative contribution of the *GLA* variant and general risk factors (e.g., smoking, diabetes) remains to be determined [6]. A similar clinical presentation is observed in male FD patients with variants that have less deleterious effects on enzyme activity. These variants are often referred to as causing late-onset, atypical, or non-classical FD. Women carrying these ‘late-onset’ variants do not develop FD symptoms and may have a normal life expectancy, free from FD-related events [7]. This means that for the evaluation of therapeutic effectiveness, these patient groups should be analyzed separately.

Importantly, there are also *GLA* variants that result in reduced AGAL enzyme activity but do not lead to FD: the residual enzyme activity is apparently sufficient to prevent substrate accumulation, which is the driver of major lysosomal dysfunction in FD [8]. The level of residual enzyme activity required to prevent substrate accumulation may differ between tissues, and the 4-methylumbelliferone (4-MU) assay generally used to determine residual enzyme activity in leucocytes lacks accuracy at very low enzyme activity levels. Therefore, with the current methods, there is no straightforward relationship between measured residual AGAL activity and disease severity [9]. What remains unclear at this time is whether carrying a ‘mild’ *GLA* variant, not resulting in abundant substrate accumulation, may contribute to the overall risk of multifactorial conditions such as coronary artery disease and cerebrovascular disease, rather than causing a classic Mendelian effect [10,11,12]. If this is the case, two situations can be identified, as follows: (1) a monogenetic disorder named FD caused by ‘severe’ *GLA* variants and (2) *GLA* variants that may be considered cardiovascular disease (CVD) risk factors, impacting health only in the presence of other CVD risk factors [8,9]. The consequence would be that only in the first situation FD-specific therapy should be considered (as well as treating other CVD risk factors). In the second instance, if the *GLA* variant is a potential additional CVD risk factor with an uncertain weight of the contribution to the overall risk, it may be more prudent to treat the well-known CVD risk factors, such as smoking, hypertension, and dyslipidemia, taking into account the costs and burden of treatment with FD-specific therapy.

FD-specific therapy, as approved by the European Medicines Agency (EMA) and the Food and Drug Administration (FDA), is available in the form of recombinant enzyme replacement therapy (ERT; agalsidase alfa (Replagal), agalsidase beta (Fabrazyme)), chaperone therapy (migalastat (Galafold)), and second-generation ERT (pegunigalsidase alfa (Elfabrio)). Biosimilars may be available locally. Alternative treatment modalities are currently being investigated, including novel small-molecule chaperones, gene and mRNA therapy, and substrate reduction therapy [13]. Despite FD-specific therapy being available for over two decades, establishing robust long-term treatment effects remains challenging, and publications give mixed messages [14,15,16].

The aim of this review is to identify the factors that interfere with reaching firm conclusions regarding treatment effectiveness in FD. In the second part, recommendations will be given to address these issues, which can help clinician scientists, drug developers, and regulators in designing future studies and outcome analyses of current treatments. We start by providing a brief overview of the characteristics of currently available FD-specific therapies. Next, we identify the factors that limit treatment effectiveness evaluation in published studies. For data on the effectiveness of ERT, we evaluated all available meta-analyses given the fact that the older studies have smaller sample sizes and durations, and discussing the individual studies would not add much to already published reviews. Since we expect outcome analyses to have benefited from growing insights in FD, for publications on ERT effectiveness in the past five years (2018–2023) we evaluated single studies with primary data. For the newer FD-specific therapies, such as migalastat and pegunigalsidase alfa, we assess phase 3 studies, comparing treatment effects to placebo or the untreated state. Note that his review exclusively focuses on currently approved FD therapies and does not address unauthorized therapies, in vitro research, or animal studies.

## 2. Fabry Disease-Specific Therapies

### 2.1. Enzyme Replacement Therapy

Enzyme replacement therapy for FD involves intravenous administration of recombinant AGAL to compensate for the deficient or dysfunctional endogenous enzyme. Agalsidase alfa (Replagal) is produced in a human cell line of fibroblast and administered at a dose of 0.2 mg/kg biweekly. Agalsidase beta (Fabrazyme) is derived from hamster ovary cells and registered in a dose of 1.0 mg/kg biweekly.

### 2.2. Chaperone Therapy

Chaperone therapy involves the use of small molecules to assist in the folding of misfolded proteins, thereby restoring proper enzyme function within cells. Migalastat (Galafold), an analog of the terminal galactosidase residue of globotriaosylceramide (Gb3), is available for patients with specific ‘amenable’ GLA variants. By selective and reversible binding to the active site of mutant forms of AGAL, migalastat stabilizes the enzyme, prevents its retention and degradation in the endoplasmic reticulum, and facilitates trafficking to the lysosomes. Amenability has been defined by the manufacturer as an increase in the mutated AGAL enzyme activity by a ≥1.2-fold over baseline and an absolute increase of ≥3% corrected for wild-type AGAL activity in the presence of 10 µmol migalastat in an assay using HEK293 cells (GLP-HEK assay) [17].

### 2.3. Second Generation Enzyme Replacement Therapy

Pegunigalsidase alfa (Elfabrio) is a novel, PEGylated, chemically modified AGAL enzyme. PEGylation of the enzyme prolongs the circulatory half-life of the enzyme to approximately 80 h compared to the ±2 h for conventional ERT [18].

## 3. Meta-Analyses of FD-Specific Therapy

Since the introduction of ERT as the first FD-specific therapy, numerous reviews have been published regarding its impact on a variety of disease symptoms and manifestations. To quantify the actual effect of a therapy, the most robust evidence is obtained through meta-analyses that analyze and compare disease outcomes between treated and untreated patients. For this reason, we reviewed all meta-analyses that have investigated ERT effects in FD (see Appendix A for search strategy). Our examination encompassed the outcome measures, the estimated treatment effects, and the challenges encountered during these analyses. We focused specifically on clinical outcomes (e.g., renal insufficiency), composite outcomes (e.g., cardiac events), and surrogate outcomes (e.g., left ventricular mass index), but excluded biomarker outcomes (e.g., change in Gb3 inclusions in tissue biopsies). This was done because there is substantial and consistent evidence for the ability of the treatments to change biomarker levels (recently reviewed in Burlina et al.) [19], while uncertainty exists regarding the effect on clinical outcomes. We will, however, in the discussion part of this review address the potential role of assessing changes in biomarkers.

### 3.1. Meta-Analyses on the Effect of Enzyme Replacement Therapy

Table 1 summarizes the findings of all relevant meta-analyses on ERT effectiveness in FD. A decade after the initial trials of ERT, the first meta-analysis by Alegra and colleagues was published, including 10 double-blind RCTs. Of all reported outcomes, only QRS complex duration could be analyzed since it was reported in a comparable fashion in different studies (Table 1) [20]. Around the same time, the meta-analysis by Rombach et al. examined the impact of agalsidase alfa and agalsidase beta on glomerular filtration rate (GFR) and left ventricular mass (LVM) measured by echocardiography in male and female FD patients separately [21]. El Dib et al. performed a meta-analysis of cohort studies only, since their preceding Cochrane review, which included RCTs only, did not yield a definitive conclusion regarding the impact of ERT on clinically relevant outcomes [15]. This meta-analysis examined several composite outcomes, including all-cause mortality, renal events (dialysis for end-stage kidney disease, kidney transplantation), cardiovascular events (myocardial infarction, cardiovascular device implantation, severe arrhythmia, congestive heart failure), and cerebrovascular events (stroke, TIA).

More recently, Sheng et al. conducted a meta-analysis investigating the effect of ERT on stroke recurrence prevention [22] and Ortiz et al. on the effect of ERT on eGFR [23]. Lee et al. explored the impact of ERT on left ventricular hypertrophy (LVH), as defined by changes in left ventricular mass index (LVMI) measured by echography [24].

### 3.2. Evaluation of Meta-Analyses on the Effect of Enzyme Replacement Therapy

The meta-analyses reported diverse effects of ERT across different patient subgroups and on a variety of outcomes. The limited availability of comparable outcome data reported in the same manner in different studies was a problem in all of them. For example, in the meta-analysis by El Dib et al., of all 77 eligible studies, fewer than 15 addressed the same outcome [15]. In the meta-analysis by Alegra et al., outcomes could not be analyzed because numerical values were rarely reported and there were differences in the units of measurement and methods of analysis [20]. Interestingly, many of the meta-analyses included multiple studies that reported on the same set of patients. Most of the studies included in these meta-analyses originated from the early 2000s, published relatively shortly after the introduction of ERT. Even the more recent meta-analyses still rely heavily on these early studies. Thus, limitations in the initial studies are reflected in subsequent meta-analyses, especially as it becomes more difficult to publish newer studies if they report on the same topic as previous studies.

The most consistent limitation across all studies was the heterogeneity in sex, age, and disease severity among the included patients [15,20,21,22,24]. The average age of the included patients in the meta-analyses ranged from their 30s to 40s [15,20,21,22,24], but from those analyses providing more detailed information, it became apparent that there was a very wide age range [20,21]. In addition, in the meta-analysis of El Dib, most studies focused on patients in whom treatment was initiated after the age of 25 [15]. The considerable variance in age at the time of treatment initiation as well as at the time of study conduction will greatly influence the observed disease severity since FD is a slowly progressive disorder. Unfortunately, clarification regarding the disease stage or the presence of comorbidities was generally not provided [15,20,22,24]. In all meta-analyses, the majority of treated patients were male [15,20,21,22,24], while groups of untreated patients predominantly included female patients [15,22,24]. The phenotypes of included patients were often not reported, but when provided, most patients exhibited the classical phenotype [15,22,24]. None of the meta-analyses specifically focused on non-classical FD. However, both Sheng and Lee noted a larger number of non-classical FD patients in the untreated patient groups of the included studies [22,24], which may indicate selection bias.

In the meta-analysis by Rombach et al., the aim was to study results per sex separately and stratified by disease stage [21]. They could not analyze treatment effects for cerebral white matter lesions (WML) development due to varying effect estimates. In addition, there was a lack of data on the incidence of clinical complications reported separately for male and female patients. El Dib proposed a subgroup sensitivity analysis based on sex, phenotype, and disease severity at therapy initiation, but this was not feasible due to insufficient reporting of this information across the included studies. Additionally, assessing the impact of follow-up duration (<5 years vs. ≥5 years) was impossible, as individual patient data were not available [15].

Ortiz et al. pursued an alternative approach by modeling a meta-analysis of the effect of ERT on eGFR using individual patient-level data (IPD) rather than group-level data [23]. They included patients with the classic disease phenotype only, leveraging IPD to adjust for observed confounders at the individual level—a crucial step for achieving balance at the group level in the face of a lack of randomized studies. However, the possibility of retrieving IPD from literature turned out to be limited. After adjusting the principal model for the imbalances in sex and proteinuria at baseline, the meta-analyses demonstrated that treated patients experienced a significantly smaller annual decrease in eGFR compared to untreated patients. Male sex and higher proteinuria levels were associated with steeper annual declines in eGFR. Despite a model with IPD, this study could not explain the heterogeneity in the annual change in eGFR.

## 4. Recent Evaluations of FD-Specific Therapies

Since FD-specific therapy became available more than 20 years ago, our understanding of complications development in FD has significantly improved. In addition to sex and *GLA* variant, the disease stage at the time of therapy initiation, which is in part determined by the patient’s age, is an important determinant of treatment outcomes [25]. Untreated plasma lysoGb3 levels are a relevant biomarker reflecting individual patients’ predicted disease development at the time of diagnosis [4]. Studies performed in the past decade have provided insight into the rate of development of FD manifestations and complications [7,26,27,28], providing insight into the study durations needed to capture specific treatment effects. We analyzed whether these insights have been implemented in more recent FD treatment studies. To this end, we searched PubMed for ERT effectiveness studies published in the last 5 years (2018–2023) (see Appendix B for search strategy). Studies were considered eligible if they reported primary data on treated patients (excluding literature cohorts or reviews) and compared outcomes with a distinct group of untreated patients (excluding cross-over studies). Case reports and series were excluded.

In total, we identified four studies that reported primary data on the effects of ERT, two studies on migalastat, and one study on pegunigalsidase alfa fitting the inclusion criteria. Open-label extension studies or switch-over trials were not included in this review because of the absence of a control group.

### 4.1. Recent Studies on the Effect of ERT (Agalsidase Alfa and Agalsidase Beta)

Table 2 summarizes the findings of recent studies with primary data on the effect of ERT in FD. Hongo et al. evaluated echocardiographic outcomes separately in 17 male and 25 female ERT-treated FD patients, with a follow-up of approximately a decade. Patients received either agalsidase alfa or beta or switched from one to another. Of all the different parameters measured, only changes in LVM indexed to height were directly comparable to data of untreated patients from a natural history description [29].

Nordin et al. utilized cardiac magnetic resonance imaging (cMRI) and cardiac biomarkers to quantify the effect of ERT on myocardial storage, inflammation, and hypertrophy after one year of therapy. A group of patients, not matched for sex, age, or phenotype, starting treatment with agalsidase alfa or agalsidase beta, was compared to a group of untreated patients and a group of long-term treated patients (not further discussed here) [30].

Van der Veen et al. performed a cross-sectional study, comparing 7 male FD patients to 23 untreated patients matched for sex, phenotype (classical FD), and age [31]. In the young classical male FD patients treated with agalsidase beta, the extent of renal, cardiac, and cerebral involvement was compared to their untreated counterparts.

Pogoda et al. employed two-dimensional speckle tracking echocardiography (2DSTE) to assess the effectiveness of agalsidase alfa and agalsidase beta (also for migalastat, not further discussed here) [32]. They analyzed yearly changes in 20 cardiac parameters during a treatment period of approximately six years. Since all untreated patients in this study were female, the comparison of the effect of ERT was limited to female FD patients.

### 4.2. Evaluation of Recent Studies on the Effect of ERT

The studies discussed had follow-up times of 5 years or longer [29,31,32], except the Nordin study, in which the follow-up duration was only one year [30]. In the Hongo study, data from treated patients were compared to untreated data from a published natural history cohort [29]. The comparison between treated and untreated patients was done for male and female FD patients separately, but the phenotype of the FD patients was not specified.

In the study by Nordin et al., age and sex distribution were unequal between groups. The ERT-treated group included older and more male FD patients with a classical disease phenotype, while the untreated control group primarily consisted of female FD patients, half of whom had non-classical *GLA* variants [30]. These female FD patients have a much lower risk of developing FD signs or symptoms and should not be used as a comparator. Compared to the untreated group, the ERT-treated group exhibited more LVH, higher baseline values of MWT, LVMI, E/e’ ratio, T1 time, and left atrial area, along with reduced GLS. These baseline values were not considered in the analyses. In addition, this study only evaluated differences in parameters within groups and did not directly compare results between the two groups.

In the study by Pogoda, the treated patients were significantly older. This age difference may have influenced the results. In addition, the milder disease course in the untreated patients may have been the reason for not treating these female FD patients [32]. Notably, ERT-treated females had higher baseline values of interventricular septum thickness (IVSd), LVMI, and plasma lysoGb3, as well as a greater number of non-sense *GLA* variants compared to untreated female FD patients. Some patients in the study Pogoda carried *GLA* variants, which have been classified as non-pathogenic in certain reports, further complicating the interpretation of the results [10,33,34]. This study did not account for the issue of multiple comparisons, necessary given the small number of patients per group.

The Van der Veen study included classical male FD patients only, and treated and untreated patients were matched for age so that the type of *GLA* variant, presence of FD features, and levels of untreated plasma lysoGb3 were similar in both groups [31].

### 4.3. Phase 3 Studies on the Effect of Migalastat

Table 3 summarizes the characteristics and results of the phase 3 studies on the effect of migalastat in FD. In the FACETS trial, a 6-month comparison between migalastat and placebo was conducted, focusing on renal, cardiovascular, and patient-reported outcomes as secondary study outcomes [35].

In the ATTRACT study, previously ERT-treated FD were randomized to receive migalastat or continue ERT (agalsidase alfa or beta) during an 18-month follow-up [36]. The evaluated annualized change rates in renal function were analyzed in a model correcting for sex and 24-h urine protein levels at baseline. The composite clinical outcome included renal, cardiac, and cerebrovascular events. In order to evaluate cardiac outcomes, they analyzed LVMI and a number of additional echocardiography parameters, including ventricular PWT, IVSd, ejection fraction, diastolic and systolic grades, and fractional shortening. In addition, they conducted a subgroup analysis for LVMI, stratified by the presence of LVH.

### 4.4. Evaluation of Phase 3 Studies on the Effect of Migalastat

In both migalastat studies, patient eligibility was determined based on the amenability of their *GLA* variant to migalastat using in vitro testing. It is important to note that the response of enzyme activity to migalastat in vitro may not fully align with the in vivo response. Specifically, the increase in enzyme activity as a response to migalastat treatment can be lower in vivo [37].

In the FACETS trial, most patients were female (64%). Baseline characteristics were comparable between the migalastat-treated and placebo groups. Less than two-third of the included patients had the classical phenotype, while most others had an unknown phenotype. One patient had the non-classical phenotype.

In the ATTRACT study, the migalastat-treated group was comparable to the ERT-treated group [36]. Both groups had previously been treated with ERT, lowering the expected magnitude and rate of potential changes. In both groups, the proportion of female FD patients was slightly higher than that of male FD patients. While the most prevalent phenotype was non-classical, the study also included patients with the A143T variant previously classified as non-disease-causing [34]. For 20% of the patients, the phenotype was listed as unknown. The wide age range at the start of the study (18 to 72 years) was not taken into account in the reference ranges of LVMI. The authors mention stratification based on sex and 24-h urine protein excretion, but details on the stratification method are lacking.

The inclusion of female patients with mild *GLA* variants in these studies is problematic since they will not develop early-onset nor severe disease symptoms. Some patients in these studies will likely never develop FD symptoms, making studying treatment effects impossible [7,26]. Male patients with non-classical FD will show significant organ involvement only later in life, and disease progression is very slow [26,38]. Thus, a maximum study duration of 18 months led to conclusions regarding the disease-modifying properties of treatments in the majority of patients included in these studies.

### 4.5. Phase 3 Study on the Effect of Pegunigalsidase Alfa

Table 3 presents a summary of the results obtained from the phase 3 BALANCE study, in which the effect of pegunigalsidase alfa therapy was compared to an active control group of patients on agalsidase beta [39]. The study aimed to evaluate the non-inferiority of pegunigalsidase alfa after 2 years of therapy, specifically focusing on the annualized change in eGFR slope in FD patients with deteriorating renal function. In the subgroup analysis, the effect was analyzed for male and female FD patients separately.

**Table 3 ijms-25-09752-t003:** Summary of phase 3 studies on the effect of migalastat and pegunigalsidase alfa in Fabry disease.

Study	Follow-Up	Analyzed FD Patients (% Male, if Combined)	Pegunigalsidase-α or Migalastat		Agalsidase-α or Agalsidase-β Treated		Outcomes	Results (IMP vs. Control) Mean ± SD/Mean (95% CI)/ %/Median [95% CI]
			Treated	Age	Untreated	Age		
Germain 2016 [35] Migalastat vs. untreated	6 mos	Combined analysis of males and females	n = 28 (32.1%)	41.5 ± 13 y/o	n = 22 (40.9%)	45.1 ± 8 y/o	- LVMI ^†^ (g/m^2^)	−0.4 ± 8.2 vs. 6.3 ± 15.3
							- eGFR ^‡^ (mL/min/1.73 m^2^)	1.8 ± 1.5 vs. −0.3 ± 1.4
							- mGFR ^#^ (mL/min/1.73 m^2^)	−1.19 ± 3.4 vs. 0.41 ± 2.0
							- Urinary protein excretion (mg/24 h)	2.2 ± 252 vs. −12.9 ± 224
Hughes 2017 [36] Migalastat vs. agalsidase alfa/ agalsidase beta	18 mos	Combined analysis of males and females	n = 36 (44.4%)	50.5 ± 2.3 y/o	n = 21 (42.9%)	46.3 ± 3.3 y/o	- eGFR ^‡^ (mL/min/1.73 m^2^)	−0.40 (−2.27, 1.48) vs. −1.03 (−3.64, 1.58)
							- eGFR ^¶^ (mL/min/1.73 m^2^)	−1.51 (−3.43, 0.40) vs. −1.53 (−4.20, 1.13)
							- mGFR ^#^ (mL/min/1.73 m^2^)	−4.35 (−7.65, −1.06) vs. −3.24 (−7.81, 1.33)
							- LVPWT ^†^ (mm)	−0.35 (−0.77, 0.07) vs. 0.029 (−0.37, 0.94)
							- IVSWT ^†^ (mm)	0.58 (−2.00, 1.40) vs. 0.37 (−0.51, 1.24)
							- LVEF ^†^ (%)	−1.07 ± 0.53 vs. −0.49 ± 1.1 ^‡‡^
							- LVMI ^†^ (g/m^2^)	−6.6 (−11.0, −2.2) vs. −2.0 [−11.0. 7.0]
							- Clinical events ^††^	29% vs. 44%

	*Subgroup analysis:*	- with LVH	n = 13 (30.8%)	*NS*	n = 5 (80%)	*NS*	- LVMI ^†^ (g/m^2^)	−8.4 (−15.7, 2.6) vs. 4.5 (−20.9, 30.0)
Wallace 2023 [39] Pegunigalsidase alfa vs. agalsidase beta	24 mos	Combined analysis of males and females	n = 52 (55.8%)	43.9 ± 10.2 y/o	n = 25 (72%)	44.3 ± 10 y/o	- eGFR ^‡^ (mL/min/1.73 m^2^)	−2.51 [−3.79, −1.24] vs. −2.16 [−3.81, −0.51]

	*Subgroup analysis:*	- males	n = 29	42.6 ± 11.5 y/o	n = 18	46.5 ± 6.9 y/o	- eGFR ^‡^ (mL/min/1.73 m^2^)	−3.44 [−5.38, −1.50] vs. −2.01 [−3.98, −0.04]
		- females	n = 23	45.6 ± 8.3 y/o	n = 7	41.7 ± 14.5 y/o	- eGFR ^‡^ (mL/min/1.73 m^2^)	−1.15 [−3.11, −0.81] vs. −2.79 [−6.28, 0.70]

SD: standard deviation, CI; confidence interval, vs: versus, mos: months, yrs: years, y/o: years old, n: number, IMP: investigation medical product, NS: not specified, LVH: left ventricular hypertrophy, LVMI: left ventricular mass index, eGFR: estimated glomerular filtration rate, mGFR: measured glomerular filtration rate, LVPWT: left ventricular posterior wall thickness diastolic, IVSWT: intraventricular septum wall thickness diastolic, LVEF: left ventricular ejection fraction. ^†^: measured by echocardiography, ^‡^: eGFR (CKD-EPI); estimated GFR by *chronic kidney disease epidemiology collaboration* equation, ^¶^: eGFR (MDRD); estimated GFR by *modification of diet in renal disease* equation, ^#^: mGFR (iohexol); measured GFR by *iohexol clearance*, ^††^: composite outcome for renal, cardiac, or cerebrovascular events (including death); for full description of events see original study, ^‡‡^: standard error of the mean.

### 4.6. Evaluation of Phase 3 Study on the Effect of Pegunigalsidase Alfa

To be included in the BALANCE study, patients had to have received at least 1 year of agalsidase beta treatment and a decline in renal function on treatment (−2 mL/min/1.73 m^2^/year). Patients were stratified by urine protein to creatinine ratio (PCR) before randomization, and eGFR results over the study period were stratified by baseline eGFR. The use of additional antiproteinuric therapy was similar between both groups and remained unchanged throughout the study. Although the included patients’ phenotype was not specified, the presence of characteristic FD features (neuropathic pain, cornea verticillata, clustered angiokeratomas) required for study inclusion suggests that the majority were classical patients, with some exceptions, for example, the inclusion of patients with the N215S variant [40].

Notably, both the experimental and control groups demonstrated stabilization of renal function compared to the pre-enrollment deterioration despite the received therapy. In other words, even in the comparator group of patients on agalsidase beta, where the therapy regimen was not changed, the on-study eGFR slope improved significantly compared to the pre-enrollment slope, which may be the result of the following factors. First, the pre-enrollment presence of deteriorating renal function was based on at least three creatinine measurements over a period of 9–18 months before study inclusion. The on-study slopes were based on no fewer than 30 measurements, using centralized and standardized methods, resulting in median eGFR slopes with much less variation. Second, the Hawthorne effect may have played a role, as patients benefit from study participation due to closer observation than in standard care, regardless of the received therapy [41]. Third, study participation may have improved overall therapy adherence, explaining the ongoing decline in plasma lysoGb3 levels in male patients on agalsidase beta during follow-up. Some male patients experienced significant declines in plasma lysoGb3 measured during the 2-year study, despite already receiving agalsidase beta for an average of 6.5 years, after which stable levels are expected [42].

## 5. Interpretation of the Results

Based on the meta-analyses and recent studies, establishing definitive conclusions about the effectiveness of FD-specific therapies across different patient subgroups remains challenging due to the lack of robust evidence on hard clinical endpoints. In classical male FD patients, ERT (with the most evidence available for agalsidase beta) has a beneficial effect in ameliorating renal complications. In these patients, progression is significantly reduced if ERT is started early in the disease course. The evidence for the impact on cardiac complications is less robust. Though treatment with ERT (again, most evidence available for agalsidase beta) has been shown to reduce the increased LVM in short-term studies, it is unclear to what extent this positively impacts long-term cardiac function and prevents the occurrence of arrythmias.

For pegunigalsidase alfa, limited available evidence suggests a similar effect of pegunigalsidase alfa compared to agalsidase beta on renal function. The effect of this treatment on other disease manifestations has not been fully assessed yet. For male and female FD patients with lower untreated plasma lysoGb3 levels compared to male patients with classical FD, robust evidence for a positive impact of ERT on disease progression and development of long-term complications is lacking. The greatest contributing factors to the lack of certainty are the heterogeneity of the disease and the lack of treatment effectiveness analyses correcting for known prognostic variables.

The same is true for migalastat, where almost all patients belong to the category with large disease heterogeneity, given the fact that the therapy can only be applied in patients with specific *GLA* variants, most of which are associated with residual enzyme activity. In this disease group, cardiac manifestations and complications are most prominent, and there is a need for establishing well-defined endpoints with a known relation to clinical outcomes.

## 6. Identified Problems and Recommendations for Future Studies

### 6.1. Problems: Comparisons of Dissimilar Groups, Lack of Accounting for Prognostic Factors, Inadequate Follow-Up, and Inappropriate Choice of Study Outcomes

The availability of RCTs in FD research is limited, and those that were placebo-controlled often had a short follow-up duration, e.g., 6 months [35,43,44,45,46,47]. Most studies evaluating FD-specific therapies rely on observational designs, frequently using natural history cohorts as untreated comparator groups. However, it is important to acknowledge that comparing current patients receiving therapy to natural history descriptions controls predating the availability of ERT [48,49,50,51,52] can pose several challenges. First, historical patient cohorts consisted primarily of male patients with classical FD, as the full phenotypic spectrum of FD was not yet recognized. Currently, the clinical presentation in female FD patients is better understood, and patients with non-classical *GLA* variants represent the majority of index patients diagnosed with FD today (unpublished data). Second, patients receive more rigorous supportive and preventive therapy nowadays, such as antiproteinuric therapy [53], antihypertensives, and cholesterol inhibitors, and the rates of tobacco use have decreased in recent decades [54]. When a lower rate of complications is observed in a current patient group receiving the investigational treatment compared to a historical untreated cohort, this difference could thus be due to a difference in disease phenotype, supportive therapy, and/or environmental risk factors between these groups. An opposite effect may be observed when currently treated patients are compared to currently untreated patients. The untreated patients are likely to have milder, later-onset disease since the reason they are not treated will be because they are in the pre-symptomatic phase of the disease or they do not require FD-specific therapy at all.

Rather than comparing treated patients to natural history cohorts, some studies use cross-over designs, comparing outcomes in the same patients consecutively treated with two types of therapies [55] or comparing pre- and post-treatment outcomes [56,57,58]. However, cross-over designs are not suitable for FD studies because of the slow disease progression, which lacks the rapid fluctuations in disease status needed for determining on-off treatment effects.

Recent studies investigating the effectiveness of ERT have revealed problems in analyzing real-world evidence. The predominantly observational studies assess the effect of therapy in patient groups that vary in age, sex, disease phenotype, and disease severity at treatment initiation. These are factors known to significantly influence the rate of disease progression and risk of complications [26]. For instance, cardiac complications tend to occur approximately ten years later in male patients with a non-classical phenotype compared to those with a classical phenotype [7]. Among female patients, only those with significantly elevated untreated plasma lysoGb3 levels have a substantial risk of developing cardiac and/or cerebral complications, though later and at a lower rate compared to male patients [4].

Additionally, the timing of treatment initiation and the extent of pre-existing manifestations influence the potential benefit of FD-specific therapy. ERT does not appear to significantly alter disease course or prevent clinical events in patients with advanced disease [25,59,60,61,62]. This presents a challenging paradox in the assessment of treatment effectiveness in FD. Those patients that are at the highest risk of clinical events within the near future (for example, the number of admissions for heart failure in two years) have often progressed to a disease stage where therapy aimed at reducing lysosomal storage (e.g., FD-specific therapy) will no longer significantly influence the disease course and cannot prevent the occurrence of these events [25,59,60,62]. Consequently, this patient group is not suitable for evaluating therapy effectiveness. Conversely, patients who commence treatment early in life, where therapy is most impactful [31,63,64], will have very low event rates in the first years or even decades of follow-up [7]. Consequently, the assessment of therapy effects necessitates long follow-up, not feasible within the timeframe of a clinical trial. Moreover, incomplete follow-up may result in erroneous comparisons of the effects between treatment modalities in non-inferiority trials. Conversely, in superiority trials, the potential beneficial effect may be missed due to an inadequate follow-up period. Therefore, surrogate outcomes are required to serve as intermediate endpoints for clinically relevant outcomes.

The validity of surrogate outcomes relies on establishing a robust relationship between surrogate outcomes and clinical events. In the context of renal disease, well-established surrogate outcomes have been identified to monitor changes in renal function, and there is extensive knowledge on the behavior of these surrogate markers over time in the general population. A global Delphi consensus on standardizing clinical outcomes for clinical trials in FD recommends renal disease using either eGFR or mGFR [65]. However, the relevance of measuring renal function as an outcome for treatment effectiveness is primarily applicable to classical male FD patients and, to a lesser extent, female FD patients with higher plasma lysoGb3, though for the latter group a much larger patient cohort is needed because of the lower event rate [4,51]. Only these patients are at risk for disease progressing towards end-stage renal disease (ESRD), meaning that male and female patients carrying non-classical *GLA* variants should not be included in renal outcome evaluations.

Establishing a reliable surrogate outcome for FD cardiomyopathy is more difficult but crucial given the higher prevalence of cardiac manifestations of FD across the different patient subgroups [7,26]. Currently, there is no consensus regarding the most relevant surrogate measure, as reflected by the great variety of cardiac outcomes assessed in the evaluated studies. Commonly used surrogate markers, such as LVM and LVMI, focus exclusively on morphological changes in the heart, measured by echocardiography or more accurately by cMRI [66,67,68]. However, LVM and LVMI do not uniformly reflect the disease course across all FD patient subgroups, and hypertrophy develops with age only in those who do develop LVH. Cardiac manifestations exhibit sex-specific variability, with males showing a strong association between the presence of LVH and the development of fibrosis [26,69]. In contrast, in female FD patients, fibrosis and ultimately the development of heart failure can occur independently of LVH [26,69]. Therefore, it is essential to recognize that LVM and LVMI solely capture the hypertrophy aspect of cardiac deterioration but not necessarily the risk for arrhythmias and diastolic heart failure, which are prevalent complications across diverse FD subgroups. Early detection of cardiomyopathy should focus on capturing subtle changes in morphological, electrophysiological, and functional cardiac markers, but it should be recognized that at this moment in time, the relationship between these markers and the subsequent development of clinical cardiac complications is not yet fully established, though progress is being made (e.g., mechanical dispersion by speckle tracking echocardiography) [70].

In summary, the expected benefit of FD-specific therapy largely depends on several individual patient characteristics, including demographic factors (sex, age), genotype (classical vs. non-classical *GLA* variants), disease severity (pre-existing manifestations), biomarker level (lysoGb3), and the presence of associated risk factors or comorbidities (e.g., cardiovascular risk profile). When evaluating the effectiveness of FD-specific therapies, it is crucial to account for and adjust for these prognostic factors. Failing to consider these may lead to biased effect estimates. In the following paragraph, we provide practical recommendations for handling these prognostic factors.

### 6.2. Recommendations

Alternative methods for analyzing data from observational studies are necessary to mitigate confounding factors resulting from an unrandomized design. Two such methods are restriction and stratification. Restriction involves studying patients with specific characteristics, resulting in a more homogenous group. For instance, Ortiz et al. restricted their analysis to classical patients [23], while Van der Veen did this for classical male patients [31], and Pietilä-Effati included only patients with a classical *GLA* variant [57,58]. Stratification, on the other hand, allows for analyzing results based on baseline characteristics, such as the presence of LVH [21,30] or the level of renal dysfunction [21]. Restriction and stratification are mainly suited to deal with a limited number of prognostic factors. To account for multiple characteristics, regression analysis is an option. Ortiz performed regression analysis on the annual change in eGFR and tested the influence of all available imbalanced known confounders [23]. However, a disadvantage of regression analysis is the substantial number of measurements and events required to establish a reliable model. Unfortunately, for clinically relevant outcomes in FD, the event rate is often low.

Therefore, we strongly advocate the use of patient matching in observational studies going forward. By creating pairs of patients with aligned prognostic factors, it is ensured that a patient starting the therapy under investigation is matched to a counterpart, receiving no treatment or a different type of treatment, with a similar predicted rate of disease progression. Any difference in disease development during follow-up can then be more reliably attributed to the initiated therapy. While untreated FD patients are the ideal controls, the scarcity of such patients necessitates alternative approaches, such as using an active comparator group. These active comparators are FD patients receiving standard of care, preferably agalsidase beta [65]. For cross-sectional studies, historical data from untreated FD patients can be used, provided they have received similar supportive therapy. The majority of historical data will consist of pre-treatment data in patients who are currently receiving FD-specific therapy. Additionally, data from recently late-diagnosed patients can be used. Given that FD is an X-linked disease, each newly diagnosed index patient brings with them a significant number of affected family members of different generations [71,72], some of whom have reached older age without ever receiving FD-specific treatment. Collecting data on these late-diagnosed untreated FD patients is very useful because each untreated patient can serve as a control in multiple matched pairs, reducing the number of patients otherwise needed as active controls. A recent re-analysis of a phase 3 RCT in another rare disease demonstrated how borrowing information from matched controls in a historical registry led to similar conclusions as the initial trial but with fewer active controls needing to refrain from therapy [73].

In the quest for precise patient matching, propensity score matching emerges as a potentially valuable approach [74]. Each patient receives a single score, reflecting the likelihood of receiving therapy based on their unique prognostic factors. By pairing a patient undergoing the therapy of interest with another FD patient possessing a similar score, we can attribute any differences in disease progression to the specific therapy. Currently, the large differences between criteria for initiating enzyme replacement therapy between different centers in the world hinder propensity score matching in FD. In addition, further research into this topic is needed to determine whether propensity scores can deal with the *point of no return* in an advanced FD, after which in many countries FD-specific therapy is no longer deemed useful for the prevention of further disease progression [31].

For new therapies, a stepwise approach to assess therapeutic effectiveness can be envisioned, potentially partially in the context of a clinical trial with surrogate endpoints, ideally followed up by the collection of real-world data [75]. Therapeutic evaluation could start with assessing the effect of reducing glycosphingolipid storage. First, by measuring its effect on plasma lysoGb3 levels as well as assessing storage in tissue biopsies. This takes approximately 6 to 12 months to show significant changes [25,76,77,78,79]. Next, more organ-specific surrogate markers can be assessed (e.g., electrographic, morphological, or functional cardiac markers), with the greatest challenge being the identification of relevant markers predicting cardiac complications (Figure 1). Once changes in both biomarkers reflecting glycosphingolipid storage and proposed organ-specific surrogate outcomes indicate potential therapeutic effectiveness in a controlled study setting, the subsequent third step involves collecting real-world evidence on the effect of therapy on FD complication rates.

The infrastructure for enhancing therapeutic development and evaluation in FD may be provided by an international FD registry (Figure 2). This disease-centered registry could facilitate international collaboration and high-quality harmonized data collection for research, drug development, and regulatory and reimbursement decision-making. Industry-independent governance should be established, and data should be accessible to different stakeholders. The registry should contain a large number and variety of patients. A core set of variables should be identified in a collaborative multi-stakeholder effort [80]. These variables should be well-defined in terms of measurement method, relevant patient group, and way of registration. Capturing important baseline characteristics that serve as prognostic factors, as well as information on potential confounding factors, including supportive therapies, is essential. Long-term follow-up will provide real-world data on the long-term (comparative) effectiveness of treatments. This registry can also be used to validate surrogate outcome parameters and capture natural history data, especially important in patients with milder phenotypes who are increasingly being diagnosed. Collaborative efforts also expedite research sample collection, enabling studies on novel (bio)markers.

To conclude, by harmonizing the FD field, the growth of treatments whose effectiveness is unclear can be stopped. It is in the interest of patients that a solid evidence base is created to substantiate comprehensive treatment guidelines, ensuring adequate and appropriate use of FD treatments in all FD phenotypes.

## Figures and Tables

**Figure 1 ijms-25-09752-f001:**
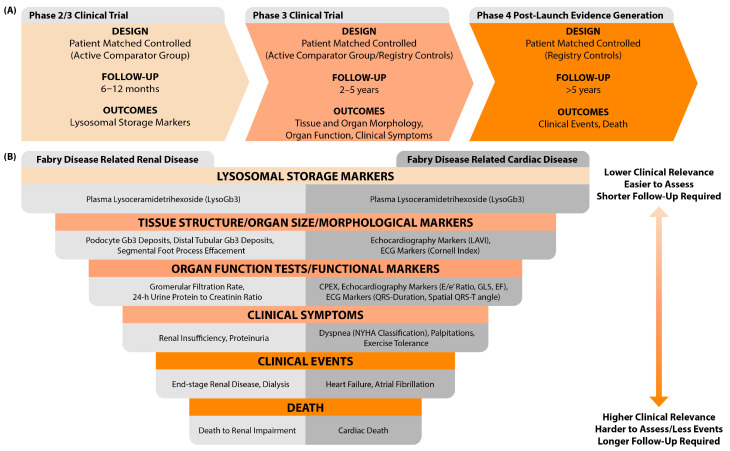
(**A**) Proposed stepwise approach for evaluating therapeutic effectiveness. In each study phase, patients initiating the investigational therapy are paired with a matched counterpart based on prognostic factors. During the initial short-term phase, this counterpart may be an active comparator (e.g., a family member). For the longer follow-up, a registry-matched counterpart is considered. Additionally, a minimum follow-up period per outcome category of interest has been suggested for each phase. (**B**) Hierarchy of clinical relevance in Fabry disease outcomes. In the figure, we recommend outcomes to evaluate therapeutic effectiveness in Fabry disease-related renal and cardiac disease. The outcomes displayed in the top layer of the figure require a shorter follow-up duration to detect changes; however, changes in these outcomes are of lower clinical relevance. For the more clinically relevant outcomes in the bottom layer, fewer events occur, necessitating longer follow-up durations. Abbreviations: Gb3 = globotriaosylceramide, LAVI = left atrial volume index, ECG = electrocardiography, CPEX = cardiopulmonary exercise testing, GLS = global longitudinal strain, EF = ejection fraction, NYHA = New York Heart Association.

**Figure 2 ijms-25-09752-f002:**
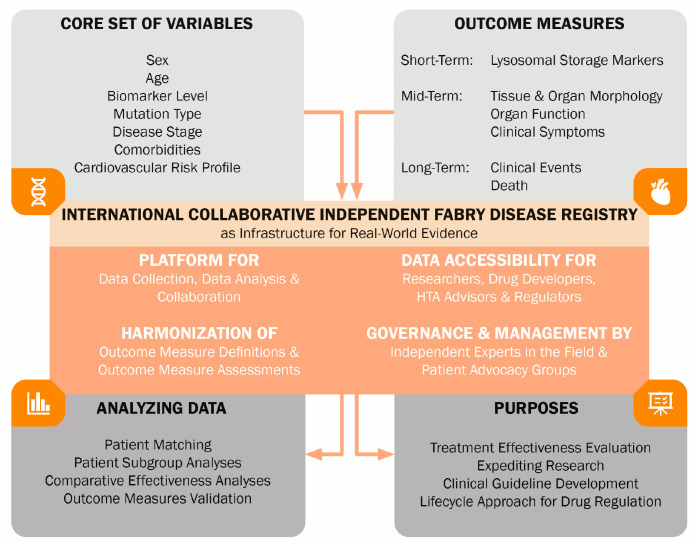
Illustration of the infrastructure of an independent international Fabry disease registry. The registry collects a standardized core set of variables and predefined outcome measures. Harmonization of these data enables structured analysis to support diverse research and clinical objectives.

**Table 1 ijms-25-09752-t001:** Summary of meta-analyses on the effect of enzyme replacement therapy (agalsidase alfa and beta) in Fabry disease.

Meta-Analysis	Included Studies (Follow-Up)	Analyzed FD Patients	FD Patients (% Male) and Number of Studies Reporting the Outcome			Outcomes	Results (Treated vs. Untreated) Mean (95% CI)/% (95% CI)/Median [IQR]
			Treated		Untreated		
Alegra 2012 [20] Agalsidase alfa/agalsidase beta vs. untreated	10 RCTs (*NS*)	Combined analysis of males and females	n = 50 (96%) 8 studies		n = 49 (100%) 8 studies	- QRS-complex duration (ms)	No statistical difference
Rombach 2014 [21] Agalsidase alfa/agalsidase beta vs. untreated	2 pooled analyses, 4 cohort studies (5 yrs)	Separate analysis of males ^†^					
		- with baseline GFR >60 mL/min/1.73 m^2^	n = 90 2 studies		n = 117 1 study	- GFR ^‡^ (mL/min/1.73 m^2^)	−2.57 (−3.21, −1.93) vs. −3.00 (−3.20, −2.80)
		- with baseline GFR <60 mL/min/1.73 m^2^	n = 23 2 studies		n = 28 1 study	- GFR ^‡^ (mL/min/1.73 m^2^)	−3.04 (−4.99, −1.09) vs. −6.80 (−9.74, −3.86) *
		- with LVH at baseline	n = 20 2 studies		n = 18 1 study	- LVM ^§^ (g/m^2.7^)	0.34 (−2.19, 2.88) vs. 6.59 (2.67, 10.51) *
		- w/o LVH at baseline (vs. *untreated patients* *with and w/o LVH*)	n = 41 2 studies		n = 39 1 study	- LVM ^§^ (g/m^2.7^)	0.94 (0.41, 1.47) vs. 4.07 (3.76, 4.38) *
		Separate analysis of females ^†^					
		- with baseline GFR >60 mL/min/1.73 m^2^	n = 52 2 studies		n = 42 1 study	- GFR ^‡^ (mL/min/1.73 m^2^)	−0.48 (−1.32, 0.36) vs. −0.90 (−2.66, 0.86)
		- with baseline GFR <60 mL/min/1.73 m^2^	n = 6 2 studies		n = 13 1 study	- GFR ^‡^ (mL/min/1.73 m^2^)	−1.37 (−2.33, −0.41) vs. −2.10 (−5.24, 1.04)
		- with LVH at baseline	n = 38 2 studies		n = 15 1 study	- LVM ^§^ (g/m^2.7^)	−1.81 (−3.41, −0.21) vs. 3.77 (−0.13, 7.67) *
		- w/o LVH at baseline (vs. *untreated patients* *with and w/o LVH*)	n = 23 2 studies		n = 39 1 study	- LVM ^§^ (g/m^2.7^)	−0.57 (−1.37, 0.23) vs. 2.31 (2.06, 2.56) *
Sheng 2019 [22] Agalsidase alfa/agalsidase beta vs. untreated	2 RCTs, 7 cohort studies (4.5 yrs)	Combined analysis of males and females	n = 1471 (62.5%) 8 studies		n = 6042 (49.1%) 5 studies	- Ischemic stroke recurrence rate	8.2% (3.8%, 12.6%) vs. 16% (10.2%, 21.7%) *
Ortiz 2020 [23] Agalsidase beta vs. untreated	2 RCTs, 8 cohort studies (2.9 ± 1.4 yrs)	Combined analysis of males and females	n = 133 (68.3%) 8 studies		n = 182 (92.9%) 3 studies	- eGFR ^¶^ (mL/min/1.73 m^2^)	−1.01 [−3.64, 1.10] vs. −3.47 [−7.74, 0.17] *
Lee 2022 [24] Agalsidase alfa/agalsidase beta vs. untreated	2 RCTs, 5 cohort studies (4.1 yrs)	Combined analysis of males and females	n = 267 (55.1%) 7 studies		n = 285 (44.6%) 7 studies	- LVMI ^§^ (g/m^2^)	−0.149 (−0.431, 0.132) ^#^
**Meta-Analyses**	**Included** **Studies** **(Follow-Up)**	**Analyzed FD** **Patients**	**FD Patients and Number of** **Studies Reporting the** **Outcome**			**Outcomes**	**Results (Agal-α vs.** **Agal-β vs. Untreated)** **% (95% CI)**
			**Agal-α**	**Agal-β**	**Untreated**		
El Dib 2017 [15] Agalsidase alfa vs. agalsidase beta vs. untreated	39 cohort studies (*NS*)	Combined analysis of males and females	n = 344 6 studies	n = 1053 2 studies	n = 812 6 studies	- All-cause mortality	9% (3%, 16%) vs. 4.4% (0.2%, 20.1%) vs. 10.8% (2.05%, 25.2%)
			n = 168 6 studies	n = 1044 1 study	n = 1698 11 studies	- Renal complications ^††^	15.3% (4.8%, 30.3%) vs. 6% (4%, 7%) * vs. 21.4% (15.2%, 28.4%)
			n = 524 4 studies	n = 1096 3 studies	n = 5854 14 studies	- Cardiovascular complications ^††^	28% (7%, 55%) vs. 7% (5%, 8%) * vs. 26.2% (14.9%, 39.4%)
			n = 461 7 studies	n = 1062 3 studies	n = 5544 15 studies	- Cerebrovascular complications ^††^	11.1% (5.8%, 17.9%) vs. 3.5% (2.4%, 4.6%) ** vs. 17.8% (12.3%, 24%)

	*Sensitivity analysis:*	Excluding children	n = 309 4 studies	n = 1053 2 studies	n = 812 6 studies	- All-cause mortality	12% (6%, 20%) vs. 4.4% (0.2%, 20.1%) vs. 10.8% (2.05%, 25.2%)
			n = 152 5 studies	n = 1044 1 study	n = 1698 11 studies	- Renal complications ^††^	16.8% (4.1%, 35.6%) vs. 6% (4%, 7%) * vs. 21.4% (15.2%, 28.4%)
			n = 426 3 studies	n = 1053 4 studies	n = 5854 14 studies	- Cardiovascular complications ^††^	35% (11%, 65%) vs. 7% (5%, 8%) * vs. 26.2% (14.9%, 39.4%)
			n = 339 5 studies	n = 1062 3 studies	n = 5544 15 studies	- Cerebrovascular complications ^††^	10.5% (4.3%, 19%) vs. 3.5% (2.4%, 4.6%) ** vs. 17.8% (12.3%, 24%)

*: statistically significant difference between the treated and the untreated group, **: statistically significant difference compared to both the agalsidase alfa treated and the untreated group, CI; confidence interval, IQR: interquartile range, vs: versus, yrs: years, n: number, agal-α: agalsidase alfa, agal-β: agalsidase beta, w/o: without, NS: not specified, LVH: left ventricular hypertrophy, LVM: left ventricular mass, LVMI: left ventricular mass index, GFR: glomerular filtration rate, eGFR: estimated glomerular filtration rate, ^†^: number of included patients not specified in meta-analysis, but retrieved from the included original studies, ^‡^: combination of measured GFR and eGFR, ^§^: measured by echocardiography, ^¶^: eGFR (CKD-EPI); estimated GFR by *chronic kidney disease epidemiology collaboration* equation, ^#^: the difference in means between the treated group and untreated group favors treatment when the value is less than 0, and favors the untreated group when the value is greater than 0, ^††^: for full description of complications see original study.

**Table 2 ijms-25-09752-t002:** Summary of recent studies with primary data on the effect of enzyme replacement therapy (agalsidase alfa and beta) in Fabry disease.

Study	Follow-Up	Analyzed FD Patients (% Male, If Combined)	Treated	Age	Untreated	Age	Outcomes	Results (Treated vs. Untreated) Mean ± SD/ Median [Range]/%
Hongo 2018 [29] Agalsidase alfa/agalsidase beta vs. untreated	11.6 ± 2.4 yrs	Separate analysis of males	n = 17	30.7 ± 9.5 y/o	n = 39	31.3 ± 11.9 y/o	- LVM ^†^ indexed to height (g/ht^2.7^/year)	1.25 ± 1.39 vs. 4.07 ± 1.03 *

	*Subgroup analysis:*	- with LVH	n = 6	*NS*	n = 18	*NS*	- LVM ^†^ indexed to height (g/ht^2.7^/year)	1.51 ± 1.32 vs. 6.59 ± 8.5
		- w/o extensive LVH	n = 14	27.8 ± 8.1 y/o	n = 39	31.3 ± 11.9 y/o	- LVM ^†^ indexed to height (g/ht^2.7^/year)	1.00 ± 1.36 vs. 4.07 ± 1.03 *
		- matched on LVM	n = 8	36.3 ± 7.6 y/o	n = 39	31.3 ± 11.9 y/o	- LVM ^†^ indexed to height (g/ht^2.7^/year)	1.59 ± 1.23 vs. 4.07 ± 1.03 *
	7.4 ± 3.0 yrs	Separate analysis of females	n = 25	40.6 ± 14.2 y/o	n = 39	36.3 ± 17.4 y/o	- LVM ^†^ indexed to height (g/ht^2.7^/year)	0.78 ± 1.23 vs. 2.31 ± 0.81 *

	*Subgroup analysis:*	- with LVH	n = 7	*NS*	n = 15	*NS*	- LVM ^†^ indexed to height (g/ht^2.7^/year)	1.54 ± 1.37 vs. 3.77 ± 7.7
		- w/o extensive LVH	n = 22	39.1 ± 14.4 y/o	n = 39	36.3 ± 17.4 y/o	- LVM ^†^ indexed to height (g/ht^2.7^/year)	0.70 ± 1.17 vs. 2.31 ± 0.81 *
		- matched on LVM	n = 16	46.5 ± 14.1 y/o	n = 39	36.3 ± 17.4 y/o	- LVM ^†^ indexed to height (g/ht^2.7^/year)	1.00 ± 1.25 vs. 2.31 ± 0.81 *
Nordin 2019 [30] Agalsidase alfa/agalsidase beta vs. untreated	1.1 ± 0.2 yrs	Combined analysis of males and females	n = 20 (35%)	49 ± 10 y/o	n = 18 (16.7%)	41 ± 12 y/o	- LVMI ^‡^ (g/m^2^)	Unchanged vs. Increased
							- Native T1 (ms)	Increased vs. Decreased
							- T2 (ms)	Unchanged vs. Unchanged
							- Troponin T (ng/L)	Unchanged vs. Increased
Van der Veen 2022 [31] Agalsidase beta vs. untreated	Cross-sectional	Separate analysis of males	n = 7	24 [14, 26] y/o	n = 23	22 [13, 27] y/o	- LVM ^†^ (g/m^2^)	80 [67, 84] vs. 94 [59, 149] *
							- LVM ^‡^ (g/m^2^)	53 [46, 59] vs. 68 [53, 99] *
							- uACR (mg/mmol)	0,4 [0, 8.8] vs. 3,7 [0, 248] *
							- eGFR ^§^ (mL/min/1.73 m^2^)	116 [92, 132] vs. 116 [46, 165]
							- LGE	0% vs. 0%
							- IRBBB	0% vs. 31%
							- Sinus bradycardia	14% vs. 52%
							- WML	14% vs. 27%
Pogoda 2023 [32] Agalsidase alfa/agalsidase beta vs. untreated	6.3 ± 3.3 yrs	Separate analysis of females	n = 24	48 [25, 71] y/o	n = 30	28 [16, 68] y/o	20 morphological and function echocardiography parameters (see Table 3 in original article)	

*: statistically significant difference between the treated and the untreated group, SD: standard deviation, vs: versus, yrs: years, y/o: years old, n: number, w/o: without, NS: not specified, LVH: left ventricular hypertrophy, LVM: left ventricular mass, LVMI: left ventricular mass index, uACR: urinary albumin creatinine ratio, eGFR: estimated glomerular filtration rate, mGFR: measured glomerular filtration rate, LGE: late gadolinium enhancement, IRBBB: incomplete right bundle branch block, WML: white matter lesions. ^†^: measured by echocardiography, ^‡^: measured by cardiac magnetic resonance imaging, ^§^: eGFR (FAS); estimated GFR by *full age spectrum* equation.

## Data Availability

No new data were created or analyzed in this study. Data sharing is not applicable to this article.

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
