# Peer review of "Establishing Treatment Effectiveness in Fabry Disease: Observation-Based Recommendations for Improvement"

_ijms, 2024, doi:10.3390/ijms25179752_

Round 1

Reviewer 1 Report

Comments and Suggestions for Authors

This is a well presented review of clinical trial evidence for disease modifying therapy for Fabry disease.  The authors describe their methodology and findings well, and go on to describe how they think the issues they have identified might be addressed in the future.  One useful addition might be a section where the authors summarise their view of the currently available clinical trial literature.  Is there actually any strong evidence that any of the high cost drugs we are currently using actually have an impact on the course of classical Fabry disease?  This would give more urgency for those designing future trials to consider the important factors they describe in their trial design.

Author Response

Comment 1: One useful addition might be a section where the authors summarize their view of the currently available clinical trial literature. Is there actually any strong evidence <…> on the course of classical Fabry disease?

Response 1: We thank the reviewer for pointing out this valuable addition to the manuscript. We have added paragraph 5 (please find below) at page 12.

“Based on the meta-analyses and recent studies, establishing definitive conclusions about the effectiveness of FD-specific therapies across different patient subgroups remains challenging, due to the lack of robust evidence on hard clinical endpoints. In classical male FD patients ERT (with most evidence available for agalsidase beta) has a beneficial effect in ameliorating renal complications. In these patients, progression is significantly reduced if ERT is started early in the disease course. The evidence for the impact on cardiac complications is less robust. Though treatment with ERT (again most evidence available for agalsidase beta) has been shown to reduce the increase LVM in short-term studies, it is unclear to what extend this positively impacts long-term cardiac function and prevents the occurrence of arrythmias. For pegunigalsidase alfa, limited available evidence suggest a similar effect of pegunigalsidase alfa compared to agalsidase beta on renal function. The effect of this treatment on other disease manifestations has not been fully assessed yet. For male and female FD patients, with lower untreated plasma lysoGb3 levels compared to male patients with classical FD, robust evidence for a positive impact ERT on disease progression and development of long-term complications is lacking. The greatest contributing factors to the lack of certainty are the heterogeneity of the disease and the lack of treatment effectiveness analyses correcting for known prognostic variables. The same is true for migalastat, where almost all patients belong to the category with large disease heterogeneity, given the fact that the therapy can only be applied in patients with specific GLA variants, most of which are associated with residual enzyme activity. In this disease group, cardiac manifestations and complications are most prominent and there is a need for establishing well-defined endpoints with a known relation to clinical outcomes.“

Reviewer 2 Report

Comments and Suggestions for Authors

In the current review the authors performed meta-analyses of papers that have evaluated clinical effectivness&therapy effects of several drugs/therapies used for Fabri Disease (FD, OMIM 301500), focusing on clinical, composite and surrogate outcomes, but not biomarker outcomes. The authors identified problems in the studies that had been performed, and formulated recommendations for the future studies (such as restriction and stratification). Across all the studies, the main limitations were use of dissimilar groups (difference/heterogeneity in age, sex, and disease severity between the study cohorts), as well as general absence of info on comorbidities or disease stage. Very reasonable is the concern expressed by the authors that in many studies treated patients were predominantly male, while untreated patients were mostly female, which might compromise the results. They also point out a possible selection bias (a larger number of patients with non-classical FD being allocated to the “untreated” groups). Next, the authors highlighted an importance of taking into account the insights on the necessary study timeframes allowing detecting delayed results of specific treatment. They also reported an inappropriate choice of study outcomes in some cases.

The main message of the review, supported by all the data summarized in this manuscript, is the unmet need 1) to establish an independent FD registry through the international collaboration and harmonization of the studies design to make results of all studies compatible (including obtaining comprehensive information on the phenotypes and genotypes of the patients enrolled in all studies, reporting data in the same manner, etc) and 2) to implement a stepwise approach for evaluating the effectiveness of novel treatments – see Figure 1 for proposed strategy. Notably, the current review focuses only on FDA-approved therapies.

Please find my comments and suggestions below.

Line 15. “GLA gene”. As per gene nomenclature guidelines, names of the genes should be italicized.

In my opinion, the sentence “Fabry disease (FD) is <> caused by mutations in the GLA gene” should be re-phrased (different wording should be used). First of all, FD is not caused by any mutations in this gene, but only by pathogenic/likely pathogenic ones. Secondly, there is a tendency now amongst the medical geneticists to adjust a vocabulary and use words “nucleotide sequence variant” (NSVs) or “variant” instead of “mutation”.

Furthermore, the major functional consequence of the presence of pathogenic/likely pathogenic NSVs in the GLA gene is an aberrant accumulation of glycosphingolipids in lysosomes, detrimental for variety of cell functions. However, the authors have not said a word about it in the abstract. 

Line 16. The authors state “The clinical phenotype of FD is highly variable, ranging from classical male patients with distinctive disease features and course to female patients with nonspecific late-onset symptoms”. Yes, there are classical and non-classical (often referred to as atypical or late onset) forms of FD, but the classical one is not exclusively observed in males even though it is most often diagnosed in men (almost male-exclusive), and female patients with classical form also exist, as well as male patients with non-classical form. Moreover, there is a staggering heterogeneity of nonspecific clinical symptoms of FD in females.

In the line 21 the authors mention several drugs used for FD, writing “this review summarizes and evaluates meta-analyses of the effectiveness of ERT and recent studies on agalsidase alfa/beta, pegunigalsidase alfa, and migalastat”, but they do not introduce them properly in the abstract (they provide all details in the line 75 although, as well as in lines 101, 107 and 118), so the readers outside of the field have to guess what the mechanisms of action of these drugs are. Notably, migalastat is an enzyme chaperone; pegunigalsidase alfa is a PEGylated recombinant enzyme for ERT, etc, so the mechanisms of their action are different. I suggest re-phrasing this sentence as well. The authors might want to say something along the lines “this review summarizes and evaluates meta-analyses of the effectiveness of current FD therapies ERT, such as <…>”

Line 60. The authors write “Importantly, there are also GLA variants that result in reduced AGAL enzyme activity but do not lead to FD: the residual enzyme activity is apparently sufficient to prevent substrate accumulation causing the major lysosomal dysfunction driving FD pathology”. I recommend adding info on the range of such residual enzyme activity.

Author Response

Comment 1: As per gene nomenclature guidelines, names of the genes should be italicized.

Response 1: Thank you for pointing this out. We have made this change throughout the manuscript wherever the GLA gene is mentioned.

Comment 2: The sentence “Fabry Disease’ is <…> caused by mutations in the GLA gene” should be rephrased.

Response 2: We agree with the reviewer that the term ‘variant’ may be more appropriate than ‘mutations’. Additionally, we acknowledge that the original wording may suggest that Fabry disease is caused by any variant found in the GLA gene, which is not accurate. To address these concerns, we have revised the text to “pathogenic GLA gene variants” in the abstract and to “GLA gene variant” throughout the manuscript.

Comment 3: The major functional consequence <…> aberrant accumulation of glycosphingolipids in lysosomes <…> is not mentioned in the abstract.

Response 3: We thank the reviewer for highlighting the omission of this important pathophysiological mechanism. We agree that this cellular consequence is important to mention, and we have accordingly added this information to the abstract at line 16 and 17.

Comment 4: The classical phenotype is not exclusively observed in males <…>, and female patients with classical form also exist, as well as male patients with non-classical form.

Response 4: We agree and apologize for any confusion caused. Our intention was not to that the classical form of Fabry disease only exists in male patients or that the non-classical form is exclusive to female patients. Our intention was to describe the broad spectrum of phenotypic expression in Fabry disease, ranging from classical male patients, who are more severely affected, to non-classical female patients, who experience milder symptoms. To clarify this distinction, we have revised the sentence accordingly at line 17, 18, and 19 in the abstract.

Comment 5: The authors mention several drugs used for FD <…>, but they do not introduce them properly in the abstracts <…>, so the readers outside of the field have to guess what the mechanisms of these drugs are.

Response 5: We would like to thank the reviewer once again for this valuable feedback. Due to the limited word count in the abstract, we initially included only the names of the drugs evaluated in this study, without mentioning their mechanisms. However, we agree that it is important for readers outside the field to understand these mechanism. Therefore, we have now included this information in the abstract at line 22 and 23.

Comment 6: The authors write “There are GLA variants that result in reduced AGAL enzyme activity but do not lead to FD: the residual enzyme activity is apparently sufficient to prevent substrate accumulation <…>”. I recommend adding info on the range of such residual enzyme activity.

Response 6: We agree with the reviewer that understanding the range of residual enzyme activity necessary to prevent pathogenic substrate accumulation would be insightful. We have carefully considered this suggestion. However, it is challenging to provide a definitive range due to the variability across different tissue types and other influencing factors, as described in Oliveira et al. 2019. In addition, the most widely used 4MU assay to measure enzyme activity in leucocytes lacks accuracy in the lower range, which is why a robust correlation between residual enzyme activity and clinical manifestations is lacking. This information has now been added to the introduction of the manuscript, starting at line 104.